# Collapse by Conditioning: Training Class-conditional GANs with Limited Data

**Mohamad Shahbazi, Martin Danelljan, Danda Pani Paudel & Luc Van Gool**
Computer Vision Lab (CVL), ETH Zurich, Switzerland
`{mshahbazi,martin.danelljan,paudel,vangool}@vision.ee.ethz.ch`

## Abstract

Class-conditioning offers a direct means to control a Generative Adversarial Network (GAN) based on a discrete input variable. While necessary in many applications, the additional information provided by the class labels could even be expected to benefit the training of the GAN itself. On the contrary, we observe that class-conditioning causes mode collapse in limited data settings, where unconditional learning leads to satisfactory generative ability. Motivated by this observation, we propose a training strategy for class-conditional GANs (cGANs) that effectively prevents the observed mode-collapse by leveraging unconditional learning. Our training strategy starts with an unconditional GAN and gradually injects the class conditioning into the generator and the objective function. The proposed method for training cGANs with limited data results not only in stable training but also in generating high-quality images, thanks to the early-stage exploitation of the shared information across classes. We analyze the observed mode collapse problem in comprehensive experiments on four datasets. Our approach demonstrates outstanding results compared with state-of-the-art methods and established baselines. The code is available at `https://github.com/mshahbazi72/transitional-cGAN`

## 1 Introduction

Since the introduction of generative adversarial networks (GANs) by Goodfellow et al. (2014), there has been substantial progress in realistic image and video generation. The contents of such generation are often controlled by conditioning the process by means of conditional GANs (Mirza & Osindero, 2014). In practice, conditional GANs are of high interest, as they can generate and control a wide variety of outputs using a single model. Some example applications of conditional GANs include class-conditioned generation (Brock et al., 2018), image manipulation (Yu et al., 2018), image-to-image translation (Zhu et al., 2017), and text-to-image generation (Xu et al., 2018).

Despite the remarkable success, training conditional GANs requires large training data, including conditioning labels, for realistic generation and stable training (Tseng et al., 2021). Collecting large enough data is challenging in many frequent scenarios, due to the privacy, the quality, and the diversity required, among other reasons. This difficulty is often worsened further for datasets for conditional training, where also labels need to be collected. The case of fine-grained conditioning adds an additional challenge for data collection, since the availability of the data samples and their variability are expected to deteriorate with the increasingly fine-grained details (Gupta et al., 2019).

While training GANs with limited data has recently received some attention (Karras et al., 2020a; Wang et al., 2018; Tseng et al., 2021), the influence of conditioning in this setting remains unexplored. Compared to the unconditional case, the conditional information provides additional supervision and input to the generator. Intuitively, this additional information can guide the generation process better and ensure the success of conditional GANs whenever its unconditional counterpart succeeds. In fact, one may even argue that the additional supervision by conditioning can even alleviate the problem of limited data, to an extent. Surprisingly, however, we observe an opposite effect in our experiments for class-conditional GANs. As visualized in Fig. 1, the class-conditional GAN trained on limited data suffers from severe mode collapse. Its unconditional counterpart, on the other hand, trained on the same data, is able to generate diverse images of high fidelity with a

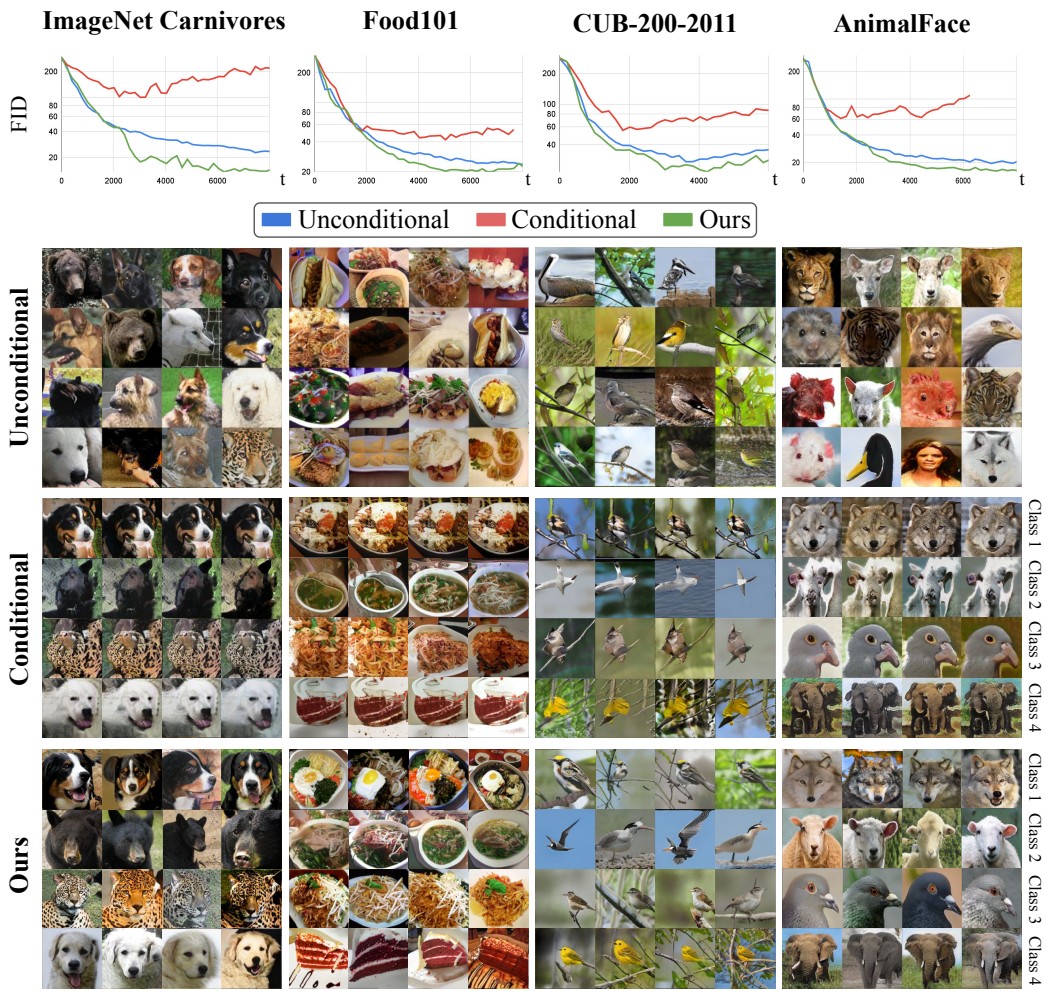

Figure 1: FID curves (first row) and sample images for training StyleGAN2+ADA unconditionally (second row), conditionally (third row), and using our method (fourth row) on four datasets under the limited-data setup (from left to right: ImageNet Carnivores, Food101, CUB-200-2011, and AnimalFace). The vertical axis of FID plots is in log scale for better visualization.

stable training process. To our knowledge, these counter-intuitive observations of class-conditional GANs have not been observed or reported in previous works.

In this paper, we first study the behavior of a state-of-the-art class-conditional GAN, with varying the number of classes and image samples per class, and contrast it to the unconditional case. Our study in the limited data regime reveals that the unconditional GANs compare favorably with conditional ones, in terms of the generation quality. We, however, are interested in the conditional case, so as to be able to control the image generation process using a single generative model. In this work, we, therefore, set out to mitigate the aforementioned mode collapse problem.

Motivated by our empirical observations, we propose a method for training class-conditional GANs that leverages the stable training of the unconditional GANs. During the training process, we integrate a gradual transition from unconditional to conditional generative learning. The early stage of the proposed training method favors the unconditional objective for the sake of stability, whereas the later stage favors the conditional objective for the desired control over the output by conditioning. Our transitional training procedure only requires minimal changes in the architecture of the existing state-of-the-art GAN model.

We demonstrate the advantage of the proposed method over the existing ones, by evaluating our method on four benchmark datasets under the limited data setup. The major contributions of this study are summarized as follows:

- We identify and characterize the problem of conditioning-induced mode collapse when training class-conditional GANs under limited data setups.

- We propose a training method for class-conditional GANs that exploits the training stability of unconditional training to mitigate the observed conditioning collapse.

- The effectiveness of the proposed method is demonstrated on four benchmark datasets. The method is shown to significantly outperform the state-of-the-art and the compared baselines.

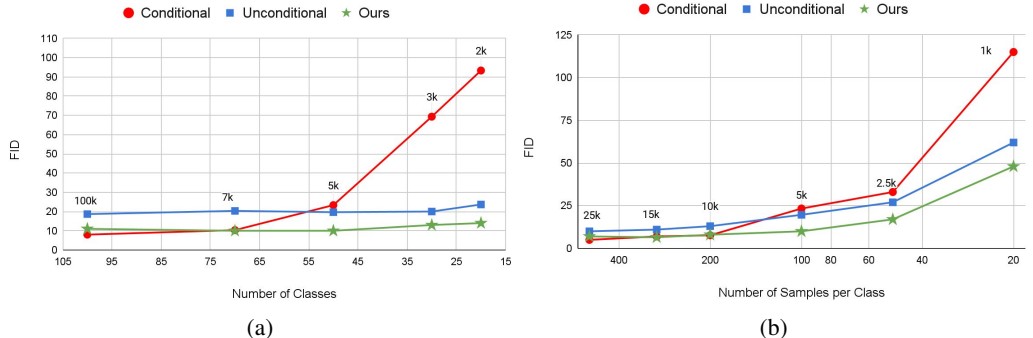

Figure 2: The FID scores for different experiments on ImageNet Carnivores using unconditional, conditional, and our proposed training of StyleGAN2+ADA by varying (a) the number of classes (number of samples per class is fixed at 100) and (b) the number of images per class (the number of classes is fixed at 50). The total number of images for the experiments is shown on the data points. The horizontal axis in Fig. (b) is in log scale for better visualization.

## 2  CLASS-CONDITIONING MODE COLLAPSE

Training conditional image generation networks is becoming an increasingly important task. The ability to control the generator is a fundamental feature in many applications. However, even in the context of unconditional GANs, previous studies suggest that class information as extra supervision can be used to improve the generated image quality (Salimans et al., 2016; Zhou et al., 2018; Kavalerov et al., 2020). This, in turn, may set an expectation that the extra supervision by conditioning must not lead to the mode collapse of cGANs in setups where the unconditional GANs succeed. Furthermore, one may also expect to resolve the issue of training cGANs on limited data, to an extent, due to the availability of the additional conditional labels.

As the first part of our study, we investigate the effect of class conditioning on GANs under the limited data setup. We base our experiments on StyleGAN2 with adaptive data augmentation (ADA), which is a recent state-of-the-art method for unconditional and class-conditional image generation under limited-data setup (Karras et al., 2020a)[1]. Both unconditional and conditional versions of StyleGAN2 are trained on four benchmark datasets (more details in Section 4.1), with the setup of this paper. The selected datasets are somewhat fine-grained, where the problem of limited data, concerning their availability and labeling difficulty, is often expected to be encountered.

In Fig. 1, we analyze the training of conditional and unconditional GANs by plotting the Fréchet inception distance (FID) (Heusel et al., 2017) during training. The analysis is performed on four different datasets, each containing between 1170 and 2410 images for training. Contrary to our initial expectation, the conditional version consistently yields worse FID compared to the unconditional one during the training. To analyze the cause, we visualize samples from the best model attained

---

[1]We originally also considered BigGAN (Brock et al., 2018) as another baseline. However, we found it to struggle with limited data in both unconditional and conditional settings.

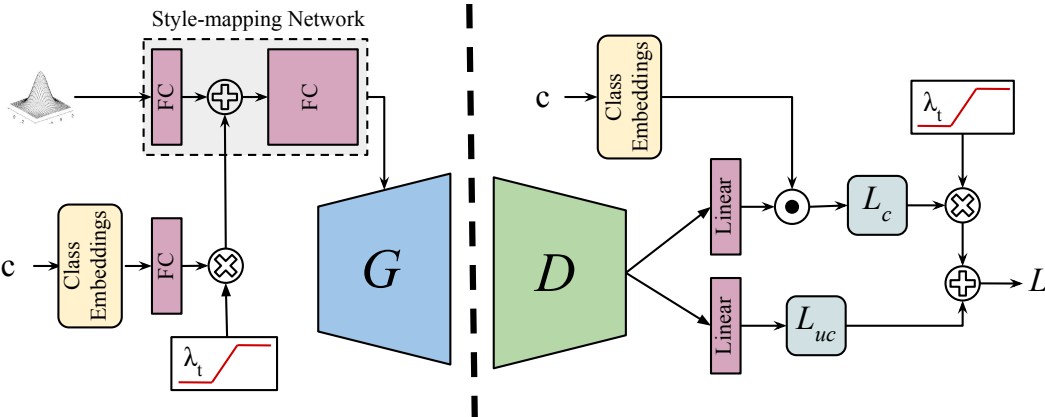

Figure 3: The proposed modified training objective and architecture of StyleGAN2 allows for transitioning from the unconditional to the conditional model during the training.

during training, in terms of FID, in Fig. 1. For each dataset, the unconditional model learns to generate diverse and realistic images, while lacking the ability to perform class-conditional generation. On the other hand, the conditional model suffers from severe mode collapse. Specifically, the intra-class variations are very small and mainly limited to color changes, while retaining the same structure and pose. Moreover, the images lack realism and contain pronounced artifacts.

Next, we further characterize the mode collapse problem observed in Fig. 1 by analyzing its dependence on the size of the training dataset. To this end, we employ the ImageNet Carnivores dataset (Liu et al., 2019), which includes a larger number of classes and images. We perform the analysis by gradually reducing the size of the training set in two ways. In Fig. 2a, we reduce the number of classes while having 100 training images in each class. In Fig. 2b, we reduce the number of images per class while using 50 classes in all cases. In both cases, the conditional GAN achieves a better FID for larger datasets, here above 5k images. This observation is in line with previous work (Salimans et al., 2016; Zhou et al., 2018; Kavalerov et al., 2020). However, when reducing the data size, the order is inverted. Instead, the unconditional model achieves consistently better FID, while the conditional model degrades rapidly.

Inspired by these observations, we set out to design a training strategy for class-conditional GANs that eliminates the mode collapse induced by the conditioning. As visualized in Fig. 1, our proposed approach, presented next, achieves stable training, leading to low FID. The images generated from our model exhibit natural intra-class variations with substantial changes in pose, appearance, and color. Furthermore, our training strategy outperforms both the conditional and unconditional models in terms of FID in a wide range of dataset sizes, as shown in Fig. 2.

## 3  METHOD

### 3.1  FROM UNCONDITIONAL TO CONDITIONAL GANS

As the analysis in Section 2 reveals, class-conditional GAN training leads to mode collapse when training data is limited, while the unconditional counterpart achieves good performance for the same number of training samples. This discovery motivates us to design an approach capable of leveraging the advantages of unconditional training in order to prevent mode-collapse in class-conditional GANs. Our initial inspections of the generated images during conditional training indicate that the mode collapse appears from the very early stages of the training. On the contrary, the corresponding unconditional GAN learns to generate diverse images with gradually improved photo-realism during training. In order to avoid mode-collapse, we aim to exploit unconditional learning at the beginning of the training process. Unconditional training in the early stages allows the GAN model to learn the distribution of the real images without the complications caused by conditioning. We then introduce the conditioning in the later stages of the training. This allows the network to adapt to conditional generation in a more stable way by exploiting the partially learned data distribution.

In general, we control the transition from unconditional training to conditional training using a transition function $\lambda_t \geq 0$. The subscript $t$ denotes the iteration number during training. Specifically, $\lambda_t = 0$ implies a purely unconditional learning, i.e. the conditional information does not affect the generator or discriminator networks. Our goal is to design a training strategy capable of gradually incorporating conditioning during training by increasing the value of $\lambda_t$. While any monotonically increasing function $\lambda_t$ may be chosen, we only consider a simple linear transition from 0 to 1 as,

$$\lambda_t = \min \left( \max \left( \frac{t - T_s}{T_e - T_s}, 0 \right), 1 \right) . \tag{1}$$

Here, $T_s$ and $T_e$ respectively denote the time steps, at which the transition starts and ends. More details on the proposed transition function are provided in the Appendix, Sec. A.1. We achieve the desired transition during training by introducing a method for controlling the behavior of the generator and the discriminator using the function $\lambda_t$. An overview of our approach, detailed in the next sections, is illustrated in Fig. 3.

## 3.2 TRANSITION IN THE GENERATOR

The generator $G(z)$ of a GAN is trained to map the latent vector $z$, drawn from a prior distribution $p(z)$, to a generated image $I_g = G(z)$ belonging to the distribution of the training data. A conditional generator $G(z, c)$ additionally receives the conditioning variable $c$ as input, aiming to learn the image distribution of the training data conditioned on $c$.

The transition from an unconditional $G(z)$ to a conditional $G(z, c)$ generator seemingly requires a discrete architectural change during the training process. We circumvent this by additionally conditioning our generator on the transition function $\lambda_t$ as $G(z, c, \lambda_t)$. More specifically, we gradually incorporate conditional information during training by using a generator of the form,

$$G(z, c, \lambda_t) = G(S(z) + \lambda_t \cdot E(c)) . \tag{2}$$

Here, $S$ and $E$ are neural network modules that transform the latent and condition vectors, respectively. In case of $\lambda_t = 0$, the conditional information is masked out, leading to a purely unconditional generator $G(S(z))$. During the transition step $T_s < t < T_e$, the importance of the conditional information is gradually increased during training.

To achieve the generator in equation 2, we perform a minimal modification to the original class-conditional StyleGAN2 (Karras et al., 2020b), where the generator consists of a style-mapping network and an image synthesis network. The style-mapping network maps the input latent vector $z$ and the embedding of the condition $c$ to the intermediate representation $w$, known as style code. The image synthesis network then generates images from the style codes. As illustrated in Fig. 3, we modify the conditioning in the generator by feeding the class embeddings to a fully-connected layer, which is then weighted by $\lambda_t$ before adding to the output of the style-mapping network's first layer.

## 3.3 TRANSITION IN THE TRAINING OBJECTIVE

The unconditional and conditional GAN frameworks also differ in their training objectives. The latter's objective assesses the realism of an image based on its conditioning variable. We propose using both unconditional and conditional objectives and follow the same transition as in the generator. Our proposed loss function includes the unconditional objective during the whole training. The conditional term, on the other hand, is weighted by the transition function $\lambda_t$. Our total objective is thus given by,

$$L^D = L^D_{uc} + \lambda_t \cdot L^D_c ,$$
$$L^G = L^G_{uc} + \lambda_t \cdot L^G_c . \tag{3}$$

Here, $L^D_{uc}$, $L^D_c$, and $L^D$ are the unconditional, conditional, and proposed losses for the discriminator, respectively. Similarly, $L^G_{uc}$, $L^G_c$, and $L^G$ represent the unconditional, conditional, and proposed losses for the generator.

As the discriminator is responsible to predict the scores needed for calculating the loss, the proposed training objective requires also modifying the discriminator's architecture. In this regard, the necessary modification must provide both conditional and unconditional scores. We propose using two

prediction branches, separately dedicated to conditional and unconditional cases, in the last layer of the discriminator, as shown in Fig. 3.

The proposed training objective and discriminator for StyleGAN2 are also visualized in Fig. 3. In the discriminator of StyleGAN2, conditioning is performed by matching the features of the input images with the target class embedding. In other words, the conditional discriminator assigns scores to the input images by calculating the dot-product between the features of the input images and the embedding of the target class $c$. The proposed discriminator architecture bears some resemblance to the architecture of projection discriminators (Miyato & Koyama (2018)). In contrast to our approach, the projection discriminator aggregates the unconditional and conditional scores inside the discriminator before the loss function. Additionally, the projection discriminator does not perform any transition between the two scores.

## 4 EXPERIMENTS

In this section, we first provide the details of our experimental setup. Then, we present the quantitative and qualitative results of the proposed method, as well as the comparison with existing methods. Finally, we provide more ablation and analysis of different components of our method.

### 4.1 EXPERIMENT SETUP

**Datasets:** We use four datasets to evaluate our method: ImageNet Carnivores (Liu et al., 2019), CUB-200-2011 (Wah et al., 2011), Food101 (Bossard et al., 2014), and AnimalFace (Si & Zhu, 2011). To keep our experiments in the limited-data regime, we decrease the number of classes and images per class in some of these datasets using random sampling.

- *ImageNet Carnivores* is a subset of the ImageNet dataset (Deng et al., 2009), which contains 149 classes of carnivore animals. The images are further processed to only contain the animal faces. We use a subset of the dataset with 20 classes and 100 images per class.
- *Food101* contains 101 different food categories, having a total amount of 101k images. We use a subset of the dataset with 20 classes and 100 images per class.
- *CUB-200-2011* contains 200 categories of different bird species with around 60 images per class. We use a subset of this dataset containing 20 classes with all the images in each class.
- *AnimalFace*, similar to ImageNet Carnivores, is a dataset that contains images of animal faces. However, the animals in AnimalFace are not limited to carnivores. AnimalFace contains 20 classes with 2432 images in total. Since AnimalFace is already a small dataset, we do not further reduce its size.

**Implementation details:** We base our method on the official PyTorch implementation of StyleGAN2+ADA. The hyper-parameter selection for the base unconditional and conditional StyleGAN2 is performed automatically as provided in the official implementation. Training is done with a batch size of 64 using 4 GPUs. For the transition function, we use $T_s = 2k$ and $T_e = 4k$ in all experiments.

**Evaluation Metrics:** We evaluate our method using Fréchet inception distance (FID), as the most commonly-used metric for measuring the similarity between the distribution of real and generated images. As FID can be biased when real data is small (Karras et al., 2020a), we also include kernel inception distance (KID) (Bińkowski et al., 2018) as a metric that is unbiased by design.

### 4.2 RESULTS AND COMPARISONS

To assess the efficacy of the proposed method, we provide a quantitative comparison with the well-established baselines and existing methods:

- *BigGAN+ADA (Brock et al., 2018)*: Achieving outstanding results on ImageNet, BigGAN has been widely used for class-conditional image generation (more details in Section 5). We use the implementation with ADA provided by (Kang & Park, 2020).
- *ContraGAN+ADA (Kang & Park, 2020)*: A class-conditional model based on BigGAN that outperforms BigGAN in many setups using self-supervision in the discriminator (more details in Section 5). We use the implementation provided by the authors.

- *U-StyleGAN+ADA:* Unconditional training of StyleGAN2+ADA using the original unconditional architecture.
- *C-StyleGAN+ADA:* Conditional training of StyleGAN2+ADA using the original conditional architecture.
- *C-StyleGAN+ADA+projD:* The modified version of conditional StyleGAN2+ADA by replacing its discriminator with the projection discriminator (Miyato & Koyama, 2018).
- *C-StyleGAN+ADA+Lecam:* Regularizing C-StyleGAN2+ADA using Lecam regularizer, recently proposed by Tseng et al. (2021) for limited-data setup (more details in Section 5). The authors have suggested a hyper-parameter in the range of $[0.1, 0.5]$. As we did not observe a noticeable difference between different values in the suggested range, we set the hyper-parameter to 0.3 for all experiments.

Results are reported in Table 1. BigGAN and ContraGAN, even though coupled with ADA, struggle to achieve any good generation quality in our experiments. Conditional StyleGAN2 shows better results on three of the datasets compared to that of the other two conditional competitors. However, the FID and KID scores are still very high. As discussed before, C-StyleGAN is consistently outperformed by its unconditional counterpart. Replacing StyleGAN2's discriminator with the projection discriminator does not yield a noticeable advantage over the original architecture, as it brings improvements on two of the datasets, but degrades the performance on the other two. Adding Lecam regularization to the C-StyleGAN2 shows promising results, achieving good FID and KID scores on Food101 and AnimalFace. However, it still fails to achieve as good generation quality as the unconditional StyleGAN2. The FID and KID scores for our proposed method indicate a significant and consistent advantage over all the compared methods in all four datasets. Our method is able to maintain a stable training and achieve better generation quality than both unconditional and conditional StyleGAN2. The FID curves during training along with the generated examples using the

Table 1: Comparison of the proposed method with baselines and existing methods on four datasets in terms of FID and KID metrics.

| Method | Carnivores | | Food101 | | CUB-200-2011 | | AnimalFace | |
|---|---|---|---|---|---|---|---|---|
| | FID | KID | FID | KID | FID | KID | FID | KID |
| BigGAN+ADA | 97 | 0.0665 | 111 | 0.0794 | 136 | 0.0860 | 90 | 0.0587 |
| ContraGAN+ADA | 97 | 0.0629 | 124 | 0.0961 | 137 | 0.0934 | 89 | 0.645 |
| UC-StyleGAN2+ADA | 23 | 0.0093 | 24 | 0.0071 | 27 | 0.0059 | 20 | 0.0048 |
| C-StyleGAN2+ADA | 100 | 0.0493 | 42 | 0.0135 | 55 | 0.0197 | 61 | 0.0107 |
| C-StyleGAN2+ADA+ProjD | 103 | 0.0503 | 32 | 0.0108 | 54 | 0.0182 | 71 | 0.0186 |
| C-StyleGAN2+ADA+Lecam | 62 | 0.0211 | 27 | 0.0086 | 37 | 0.0179 | 26 | 0.0042 |
| Ours | **14** | **0.0021** | **20** | **0.0045** | **22** | **0.0032** | **16** | **0.0018** |

proposed method are visualized in Fig. 1. FID curves indicate training dynamics as stable as the unconditional training while achieving better FID. In addition, the generated images of our method are clearly of more diversity and quality compared to those of the standard conditional model, showing the advantage of the proposed method. The results in Fig. 2 further demonstrate a clear advantage of our method over a wide range of data sizes. Our approach maintains the performance of cGANs for larger datasets, while significantly outperforming the unconditional counterpart when data is more scarce. This shows that our method enables cGANs to use the additional label information to achieve better generation quality without falling into the mode collapse induced by conditioning.

## 4.3 ABLATION AND ANALYSIS

In this section, we provide further ablation and analysis over different components of our method. First, we provide an ablation study containing four different variants:

- *No transition:* Training the modified architecture with the new objective, without any transition in the objective or the generator (equivalent to using an auxiliary unconditional loss term to train a conditional model).
- *Transition only in G:* Performing the transition only in the generator (Sec. 3.2), while the training objective is the summation of the unconditional and conditional term.

- *Transition only in loss:* Performing the transition only in the training objective (Sec. 3.3), while the generator is fully conditional from the beginning.
- *Final method:* The final method with all the proposed components.

Table 2 presents the results of the ablation study on Food101 and ImageNet Carnivores. The *No transition* version yields poor results, showing that the mode collapse is not alleviated by only adding an auxiliary unconditional training objective. Adding the transition to the generator already brings significant improvement to the model. Having the transition only in the objective, on the other, does not lead to good results. To our initial surprise, this reveals that transitioning in the generator is a crucial part of the method. However, transitioning in the objective

Table 2: Ablation study over different components of the proposed method, including the proposed architecture and objectives, as well as the transition in the generator and in the objective.

|  | Food101 | | ImageNet Carniv. | |
|---|---|---|---|---|
| Experiment | FID | KID | FID | KID |
| No transition | 79 | 0.0300 | 110 | 0.0436 |
| Transition only in G | 25 | 0.0064 | 17 | **0.0019** |
| Transition only in loss | 80 | 0.0297 | 107 | 0.0539 |
| Final method | **20** | **0.0045** | **14** | 0.0021 |

in addition to that in the generator, as proposed in our final method, achieves the best results.

Next, we analyze the impact of *when* the transition between unconditional and conditional learning is applied. The total transition time is fixed to $T_e - T_s = 2k$ time steps. We then report the results on the Food101 and ImageNet Carnivores datasets for different starting times $T_s$ in Table 3. Importantly, we notice significantly worse results if the transition is started at the beginning of the training $T_s = 0$. This further supports the hypothesis that conditioning leads to mode collapse in the early stages of the training. By introducing conditional in-

Table 3: Analysing the importance of the transition starting time ($T_s$). The transition period is constant at $2k$ for all the experiments.

|  | Food101 | | ImageNet Carnivores | |
|---|---|---|---|---|
| Experiment | FID | KID | FID | KID |
| $T_s = 0$ | 24 | 0.0068 | 27 | 0.0075 |
| $T_s = 1k$ | 22 | 0.0057 | 15 | 0.0023 |
| $T_s = 2k$ | 20 | 0.0045 | 14 | 0.0021 |
| $T_s = 4k$ | 21 | 0.0056 | 14 | 0.0021 |
| $T_s = 6k$ | 23 | 0.0056 | 15 | 0.0028 |

formation in a later stage, good FID and KID numbers are obtained without being sensitive to the specific choice of $T_s$. In Table 4, we further independently analyze the transition end time $T_e$, while keeping $T_s = 2k$ fixed. Again, our approach is not sensitive to its value. Our method, therefore, does not require extensive hyper-parameter tuning.

Lastly, we visualize the evolution of the generated images and the formation of classes during our training process. Fig 4 shows how images generated from the unconditional phase of training on AnimalFace start to evolve into different images of the class Panda. In addition to the formation of the classes, Fig. 4 shows how the image quality continues to improve during and after the transition. In the Appendix, more ablation studies (Sec. A.6, A.7), as well as im-

Table 4: Analysing the importance of the transition ending time ($T_e$). The starting time ($T_s$) is constant at $2k$ for all the experiments.

|  | Food101 | | ImageNet Carnivores | |
|---|---|---|---|---|
| Experiment | FID | KID | FID | KID |
| $T_e = 3k$ | 21 | 0.0053 | 16 | 0.0032 |
| $T_e = 4k$ | 20 | 0.0045 | 14 | 0.0021 |
| $T_e = 5k$ | 23 | 0.0062 | 15 | 0.0029 |

ages generated with our method (Sec. A.8) are provided for further assessment of the proposed method.

## 5 RELATED WORK

**Class-conditional GANs:** The first conditional GAN architecture, introduced by Mirza & Osindero (2014), incorporated conditioning by concatenating the condition variable to the input of the generator and the discriminator. AC-GAN (Odena et al., 2017) equipped the discriminator with an auxiliary classification task to ensure the conditional generation. cGAN with projection discriminator (Miyato & Koyama, 2018) proposed a new discriminator architecture, ensuring the class conditioning by computing the dot-product between image features and class embeddings. Improving over cGAN with projection discriminator, BigGAN (Brock et al., 2018) was able to become the state-of-the-art cGAN on ImageNet dataset (Deng et al., 2009). Inspired by the recent progress in self-supervised learning, ContraGAN (Kang & Park, 2020) improved over BigGAN by exploit-

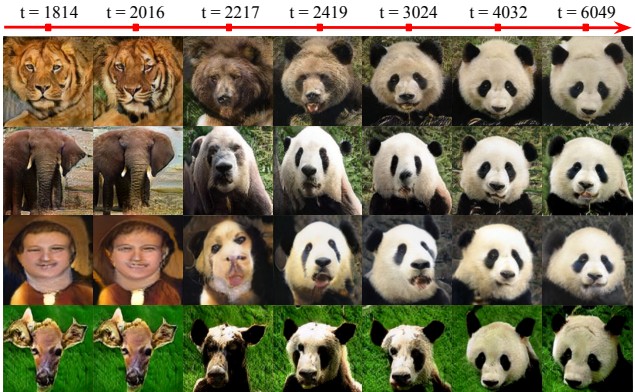

Figure 4: Visualization of the formation of the class "Panda" in AnimalFace during the transition from unconditional to conditional training. The transition starts at $t = 2k$.

ing an auxiliary self-supervised task in the discriminator for better image representation learning. StyleGAN (Karras et al., 2019), mainly known for unconditional image generation, was extended to class conditioning in the improved version, and coupled with adaptive differentiable augmentation (ADA), outperformed the state-of-the-art cGANs in small data setup (Karras et al., 2020a).

**cGANs in small data regimes:** There exist several directions to address the problem of training GANs on small data. Transfer learning (TL) exploits large pre-training data to provide better initialization for learning the target data. Several works recently have studied the best practices for TL in GANs (Wang et al., 2018; 2020; Mo et al., 2020; Zhao et al., 2020a; Noguchi & Harada, 2019)). As an example, cGANTransfer (Shahbazi et al., 2021) proposed class-specific knowledge transfer in class-conditional GANs by learning the target class embeddings as a linear combination of the ones in the pre-trained model. Although effective, TL usually requires large pre-training data with sufficient domain relevance to the target. Data augmentation (DA) is another technique for addressing small data. To prevent DA from leaking to the generated images, recent works proposed differentiable DA (Zhao et al., 2020b; Karras et al., 2020a). Adding adaptive differentiable augmentation (ADA) to StyleGAN2 resulted in a significant improvement in conditional generation from small datasets, outperforming previous models on CIFAR10 (Karras et al., 2020a). In addition to the aforementioned methods, there are other studies focusing on better architecture or objective design for small data regimes. Liu et al. (2021) proposed a lighter unconditional architecture and a self-supervised discriminator for StyleGAN. Tseng et al. (2021) proposed the Lecam regularization to prevent the discriminator from over-fitting on small data by penalizing the current difference between real and fake predictions from previous fake and real predictions, respectively.

**Conditioning collapse in GANs:** Previous studies have reported a decrease in diversity in tasks with strong pixel-level conditioning, such as semantic masks or images (e.g. super-resolution) (Ramasinghe et al., 2021; Lugmayr et al., 2022; Lee et al., 2019; Isola et al., 2017). Such lack of diversity is generally considered to be due to the conflict between the adversarial and reconstruction losses common in image-conditional GANs. The same effect, however, not only is not explored for the class-conditional setting but also does not directly translate to this setup. In this work, we discover the conditioning collapse for class-conditional GANs to occur when training data is small (Fig. 2)

## 6  CONCLUSIONS

In this work, we studied the problem of training class-conditional generative adversarial networks with limited data. Our empirical study demonstrated that class-conditioning can lead the training of GANs to mode-collapse within the investigated setup. To prevent such collapse, we presented a method of injecting the class conditioning by transitioning from unconditional to the conditional case, in an incremental manner. To enable such transition, we have proposed architectural modifications and training objects, which can be easily adapted by any existing GAN model. The proposed method achieves outstanding results compared to the state-of-the-art methods and established baselines, for the limited data setup of four benchmark datasets. In the future, we will study our method for other types of conditioning (e.g. semantics and images), as well as other architectures.

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

## A APPENDIX

In the following, we provide more details, experiments, and visualizations on the discovered observation and the proposed method.

### A.1 THE TRANSITION FUNCTION

In Fig. 5, we provide a detailed visualization of the proposed transition function. As shown, $T_s$ is the starting time and $T_e$ is the end time of the transition. Note that the end of the training is different from the end of the transition. We have added a new term $T_m$ to Fig. 7 to represent the maximum training iterations. First, the model is trained unconditionally from $t = 0$ until $t = T_s$. At $T_s$, the transition to the condition model starts, $\lambda_t$ going from 0 to 1 linearly. After the end of the transition ($T_e$), the transition function $\lambda_t$ remains at its maximum value of 1. In other words, the weight of each loss (of Equation 3 in the paper) does not get adjusted anymore. The end of the transition $T_e$ happens at about half the total training time $T_m$ in our experiments.

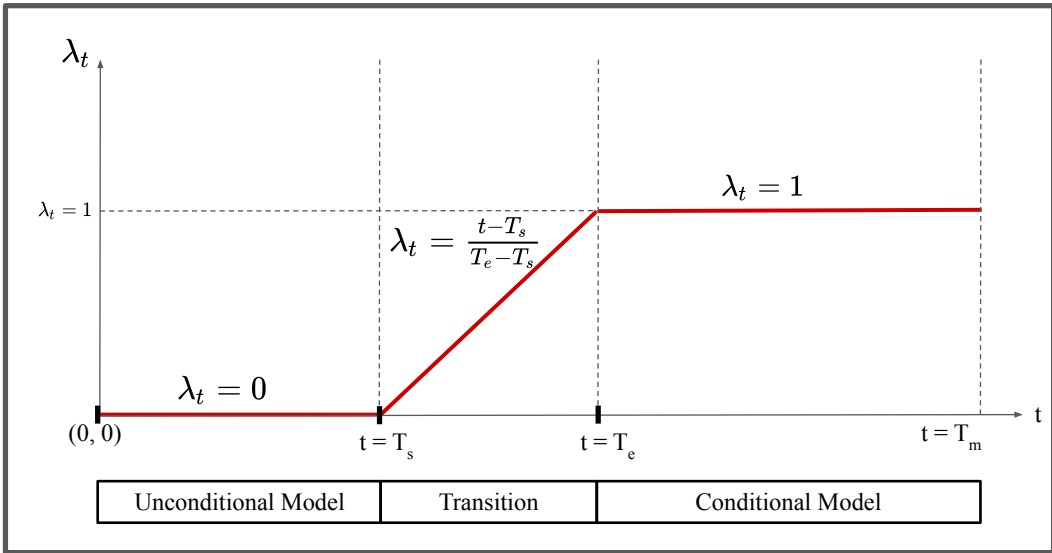

Figure 5: Visualization of the transition function $\lambda_t$. $T_s$, $T_e$, and $T_m$ denote the start of the transition, the end time of the transition, and the end of the training, respectively.

### A.2 MORE DETAILS ON THE IMPLEMENTATION

The proposed transition function makes a transition from 0 to 1 in the specified transition time. However, in the specific case of the AnimalFace dataset, we found that clipping the output of the transition function to the maximum value of 0.2 achieves the best results, which are reported in the main paper.

### A.3 EXPERIMENTS ON CIFAR100

In addition to the experiments on the four datasets presented in the paper, we provide the results of training unconditional and conditional StyleGAN2+ADA, as well as our method, on four different subsets of CIFAR100 (Krizhevsky (2009)) in Table 5. In Fig. 6, as an example, the FID curves of training the three methods on a subset of CIFAR100 with 100 classes and 300 images per class are visualized. We observe a similar behavior on CIFAR100, where our approach outperforms the conditional baseline, achieving FID and KID better or on-par with the unconditional baseline. .

Table 5: Quantitative results for examples of unconditional and conditional training of Style-GAN2+ADA, as well as our method, on different subsets of CIFAR100. In the name of the columns, C indicates the number of classes and S shows the number of images per class

| Method | C20, S500 | | C50, S50 | | C50, S300 | | C100, S300 | |
|---|---|---|---|---|---|---|---|---|
| | FID | KID | FID | KID | FID | KID | FID | KID |
| UC-StyleGAN+ADA | **7** | **0.0006** | **20** | 0.0022 | **6** | **0.0008** | **6** | 0.0021 |
| C-StyleGAN+ADA | 12 | 0.0033 | 23 | 0.0036 | 9 | 0.0025 | 13 | 0.0053 |
| Ours | **7** | **0.0006** | **20** | **0.0014** | **6** | **0.0008** | **6** | **0.0012** |

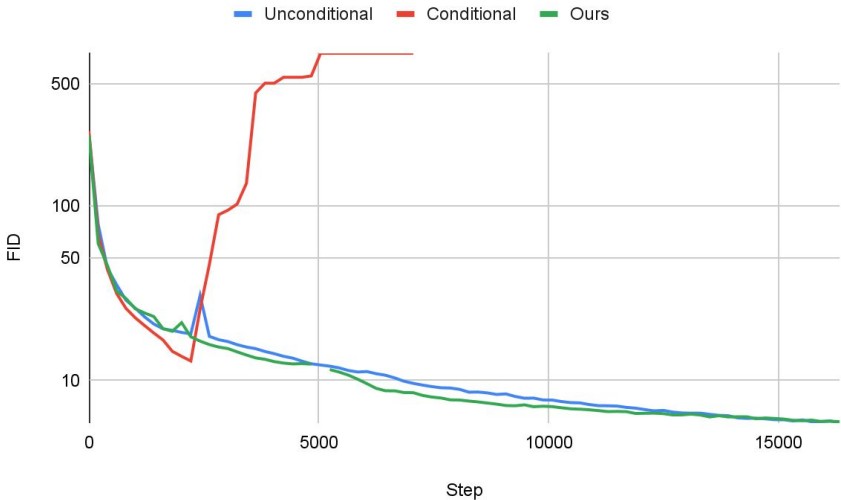

Figure 6: FID curves for training unconditional and conditional StyleGAN2, as well as our method, on CIFAR100 with 100 classes and 300 images per class. The vertical axis is in the log scale.

## A.4 CLASS-WISE FID AND KID

Following the common practice (StyleGAN, BigGAN, etc.), we calculate the FID and KID metrics reported in our experiments unconditionally, with a class sampling distribution that matches the class distribution of the training dataset. We did not provide the class-wise FID and KID due to the insufficient class-wise sample size. Here in Table 6, we report the class-wise metrics for ImageNet Carnivores and Food101, by using all the images of corresponding classes, including the additional images not used for training.

Table 6: The class-wise FID and KID for Imagenet Carnivores and Food101 using the full number of real samples per class in the evaluation.

| Loss Formulation | Carnivores | | Food101 | |
|---|---|---|---|---|
| | FID | KID | FID | KID |
| C-StyleGAN+ADA | 139 | 0.179 | 100 | 0.079 |
| C-StyleGAN2+ADA+ProjD | 151 | 0.199 | 97 | 0.0815 |
| C-StyleGAN2+ADA+Lecam | 90 | 0.096 | 56 | 0.027 |
| Ours | **30** | **0.011** | **44** | **0.019** |

Note that the values are generally larger for all methods, since a large fraction of FID/KID reference sets were not used for training. However, the relative values are still consistent with the metrics reported in the paper. These measures, along with the generated images provided later in the appendix (Figs 9-12), show the class consistency of the proposed method.

A.5   PRECISION AND RECALL

In Table 7, we provide the precision and recall proposed by Kynkäänniemi et al. (2019), with the implementation provided by StyleGAN2+ADA. Based on the results, unconditional training always yields a higher recall, as it can generate between-mode images resulting in bigger diversity. Among the conditional methods, our method yields significantly better recall, while being comparable in terms of precision. Low recall values for the conditional baselines confirm the observed mode collapse. In Table 8, we also provide the class-wise precision and recall for ImageNet Carnivores and Food101, calculated in the same manner as the class-wise FID and KID in Section A.4.

Table 7: The unconditional precision and recall for the compared methods in the paper.

| Method | Carnivores | | Food101 | | CUB-200-2011 | | AnimalFace | |
| --- | --- | --- | --- | --- | --- | --- | --- | --- |
| | Pr | Rl | Pr | Rl | Pr | Rl | Pr | Rl |
| UC-StyleGAN2+ADA | 0.77 | 0.300 | 0.71 | 0.187 | 0.70 | 0.232 | 0.76 | 0.430 |
| C-StyleGAN2+ADA | 0.80 | 0.0008 | 0.74 | 0.004 | **0.79** | 0.002 | 0.78 | 0.032 |
| C-StyleGAN2+ADA+ProjD | 0.81 | 0.0 | 0.74 | 0.002 | 0.76 | 0.067 | **0.83** | 0.0 |
| C-StyleGAN2+ADA+Lecam | 0.81 | 0.046 | 0.73 | 0.004 | 0.76 | 0.097 | **0.83** | 0.0005 |
| Ours | **0.82** | **0.263** | **0.82** | **0.068** | 0.77 | **0.229** | **0.83** | **0.314** |

Table 8: The class-wise precision and recall for the compared methods on Imagenet Carnivores and Food101 using all of the available real samples per class in the evaluation.

| Method | Carnivores | | Food101 | |
| --- | --- | --- | --- | --- |
| | Pr | Rl | Pr | Rl |
| C-StyleGAN2+ADA | **0.75** | 0.0 | 0.75 | 0 |
| C-StyleGAN2+ADA+ProjD | 0.68 | 0.0 | **0.78** | 0.00005 |
| C-StyleGAN2+ADA+Lecam | 0.67 | 0.0001 | 0.70 | 0.0148 |
| Ours | 0.73 | **0.1205** | 0.64 | **0.1049** |

A.6   MORE ABLATION: STYLEGAN2 WITHOUT ADA

To investigate whether the occurrence of the conditional collapse and efficacy of the proposed method in solving it is related to the adaptive differentiable augmentation (ADA), we perform further experiments on a subset of ImageNet carnivores (50 classes, 500 mages per class), without using ADA in the training. As the FID curves in Fig. 7 and FID and KID scores in Table 9 show, the observed conditioning collapse happens even in the absence of ADA. Our method is still able to solve the problem by leveraging the stable behavior of unconditional training

Table 9: The FID and KID scores for training unconditional and conditional StyleGAN2, as well as our method, without using ADA, on ImageNet Carnivores with 50 classes and 500 images per class.

| Method | FID | KID |
| --- | --- | --- |
| U-StyleGAN2 | 99 | 0.0795 |
| C-StyleGAN2 | 13 | 0.0067 |
| Ours | **8** | **0.0020** |

A.7   MORE ABLATION: TRANSITION IN THE LOSS FUNCTION

In Table 10, we provide the ablation over two choices for transition in the proposed loss function:

- $L = (1 - \lambda_t) \cdot L_{uc} + \lambda_t \cdot L_c$.

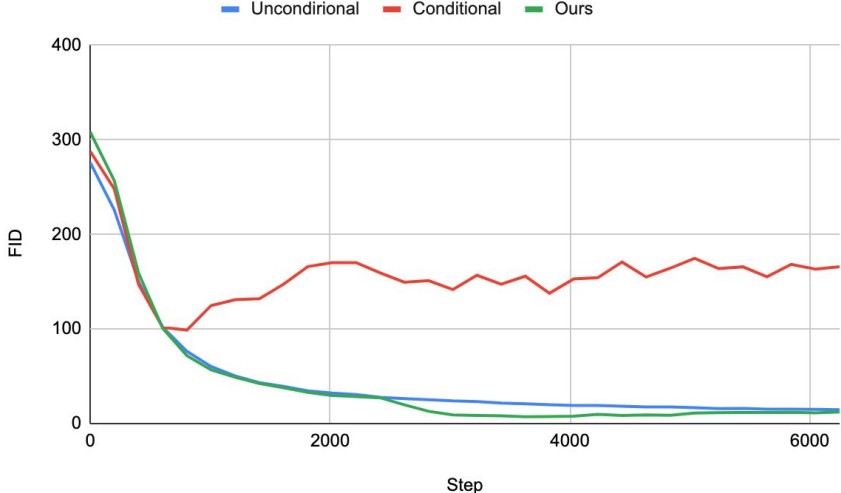

Figure 7: FID curves for training unconditional and conditional StyleGAN2, as well as our method, on ImageNet Carnivores with 50 classes and 500 images per class.

- $L = L_{uc} + \lambda_t \cdot L_c$ (Proposed in the paper).

The results show better performance for the loss formulation proposed in the paper. Based on the results, the unconditional and conditional training are not conflicting in the later stages of the training. Instead, having the unconditional loss helps the performance. However, as shown in Table 2 of the paper, the two losses seem to be conflicting in the beginning of the training, resulting in a bad performance in the absence of the transition.

Table 10: The FID and KID results for two different loss formulations.

| Loss Formulation | Carnivores | | Food101 | | CUB-200-2011 | | AnimalFace | |
|---|---|---|---|---|---|---|---|---|
| | FID | KID | FID | KID | FID | KID | FID | KID |
| $L = (1 - \lambda_t) \cdot L_{uc} + \lambda_t \cdot L_c$ | 18 | 0.0047 | 25 | 0.0088 | 25 | 0.0064 | 20 | 0.0035 |
| $L = L_{uc} + \lambda_t \cdot L_c$ (Ours) | **14** | **0.0021** | **20** | **0.0045** | **22** | **0.0032** | **16** | **0.0018** |

## A.8 VISUAL RESULTS

In Fig. 8, We provide visual results for comparison of our method and the baselines. Moreover, more images Randomly generated using our method on the four datasets are visualized in Figs. 9-12.

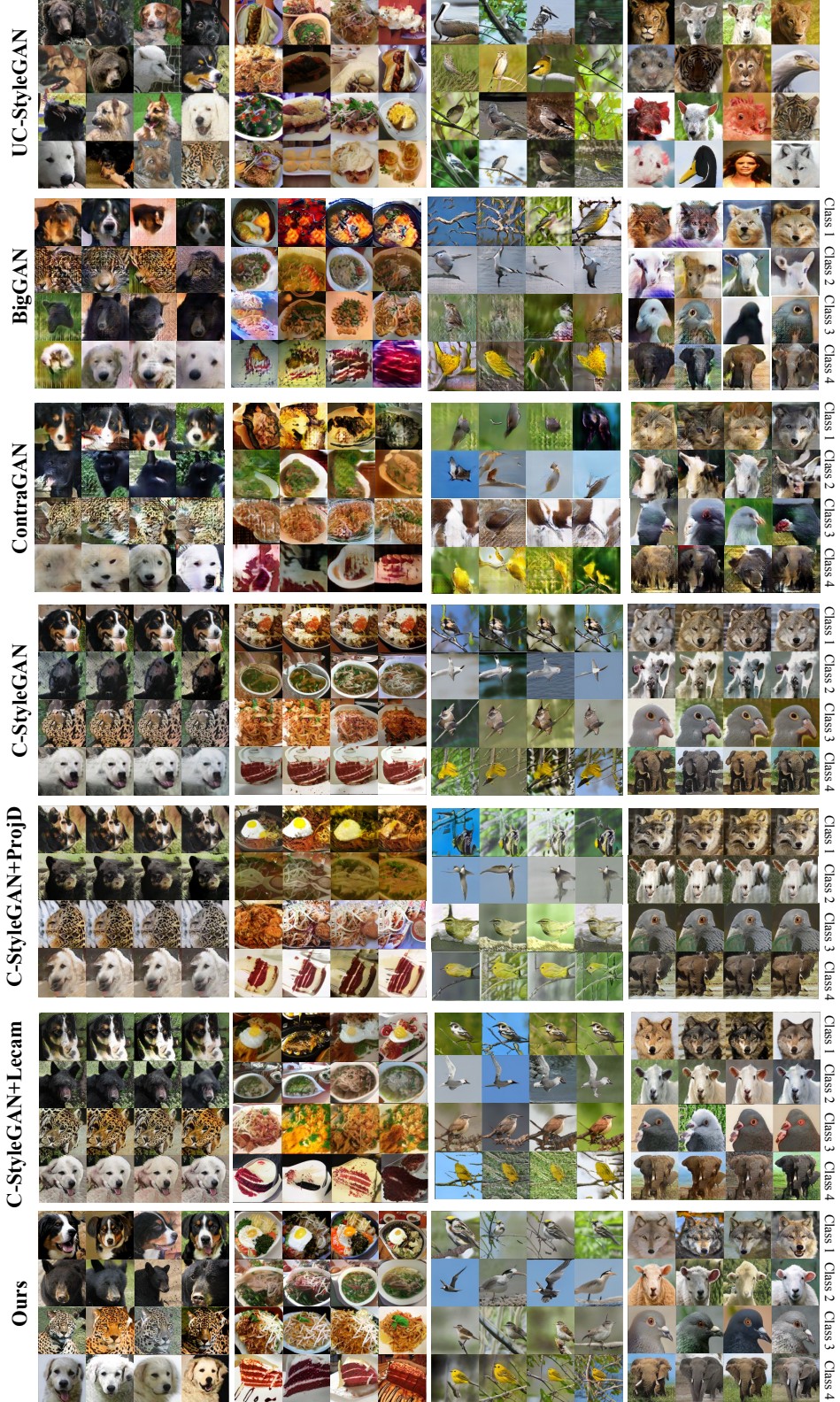

Figure 8: Visual results for the compared methods on four datasets.

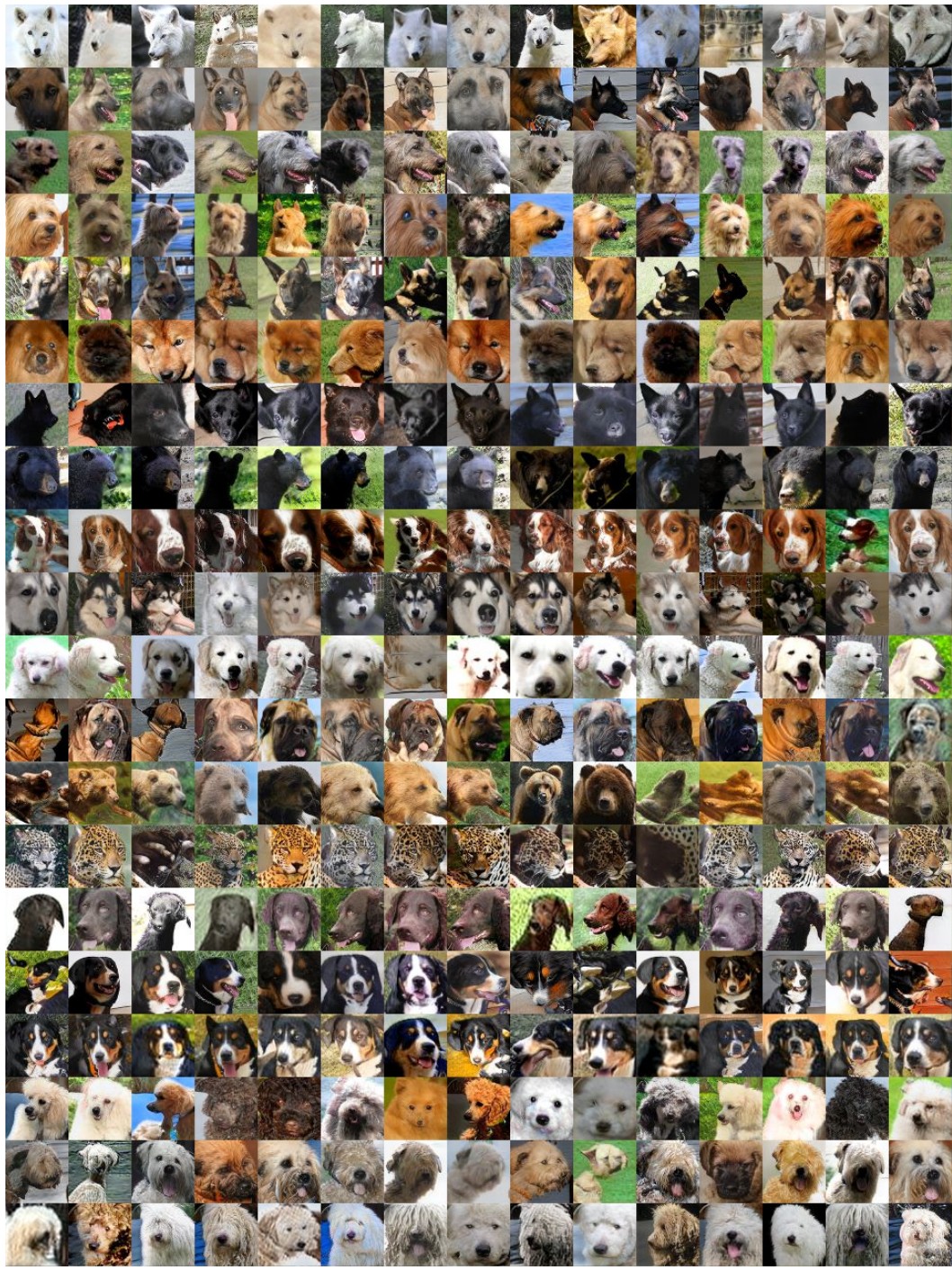

Figure 9: Randomly-generated images using the proposed method trained on ImageNet Carnivores dataset. Each row represents a different class. FID score is 14.

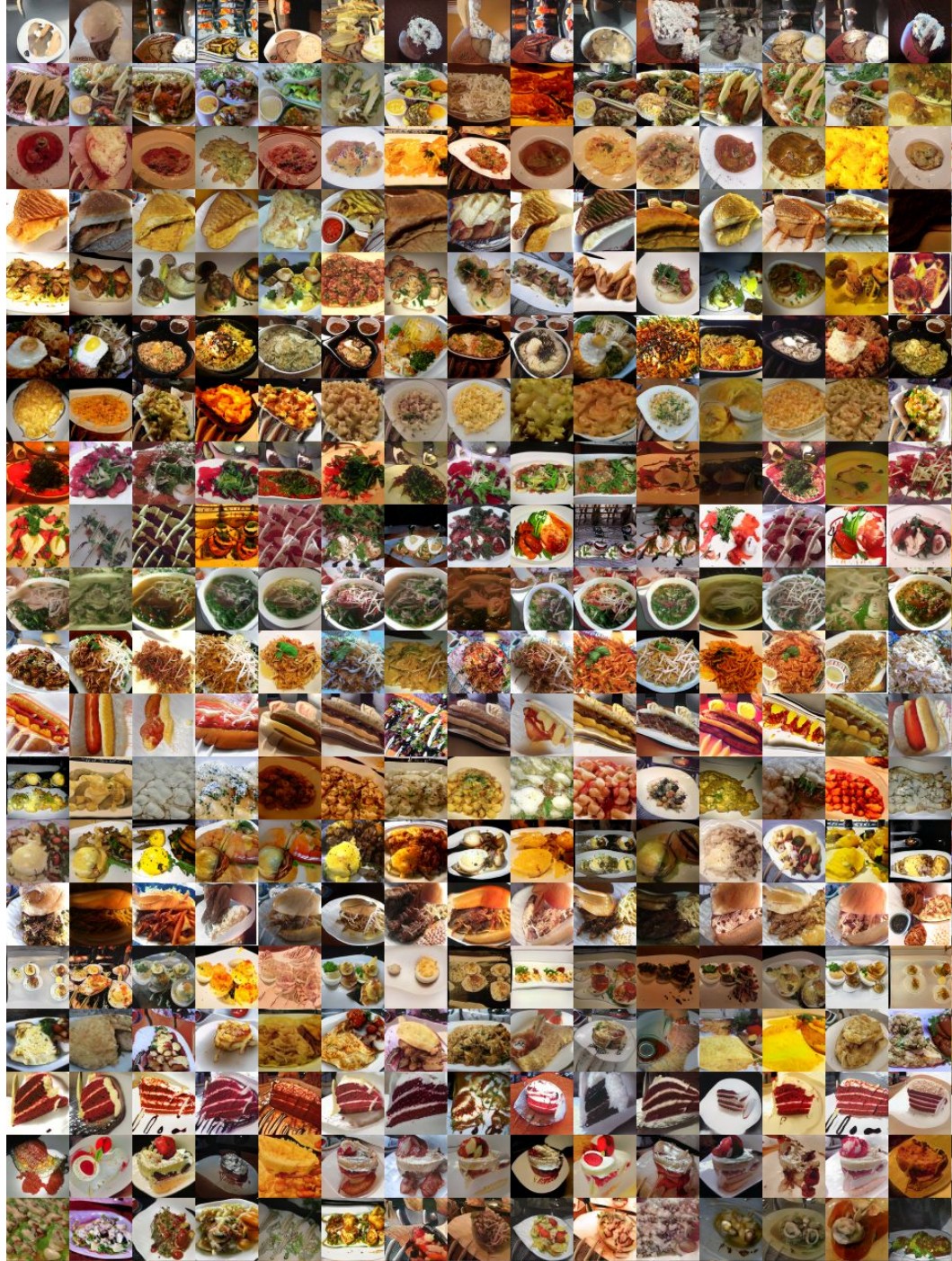

Figure 10: Randomly-generated images using the proposed method trained on Food101 dataset. Each row represents a different class. The FID score is 20.

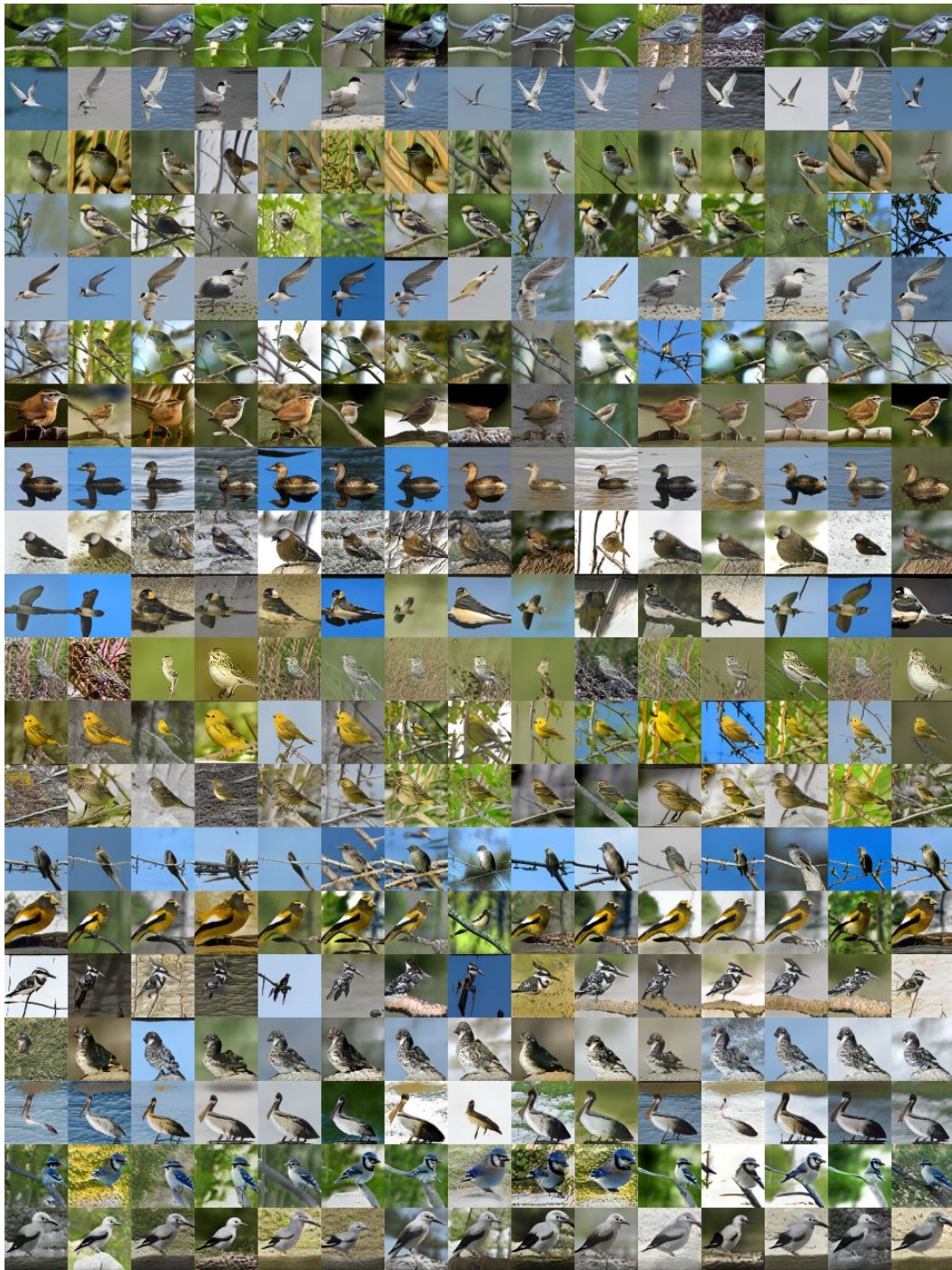

Figure 11: Randomly-generated images using the proposed method trained on CUB-200-2011 dataset. Each row represents a different class. The FID score is 22.

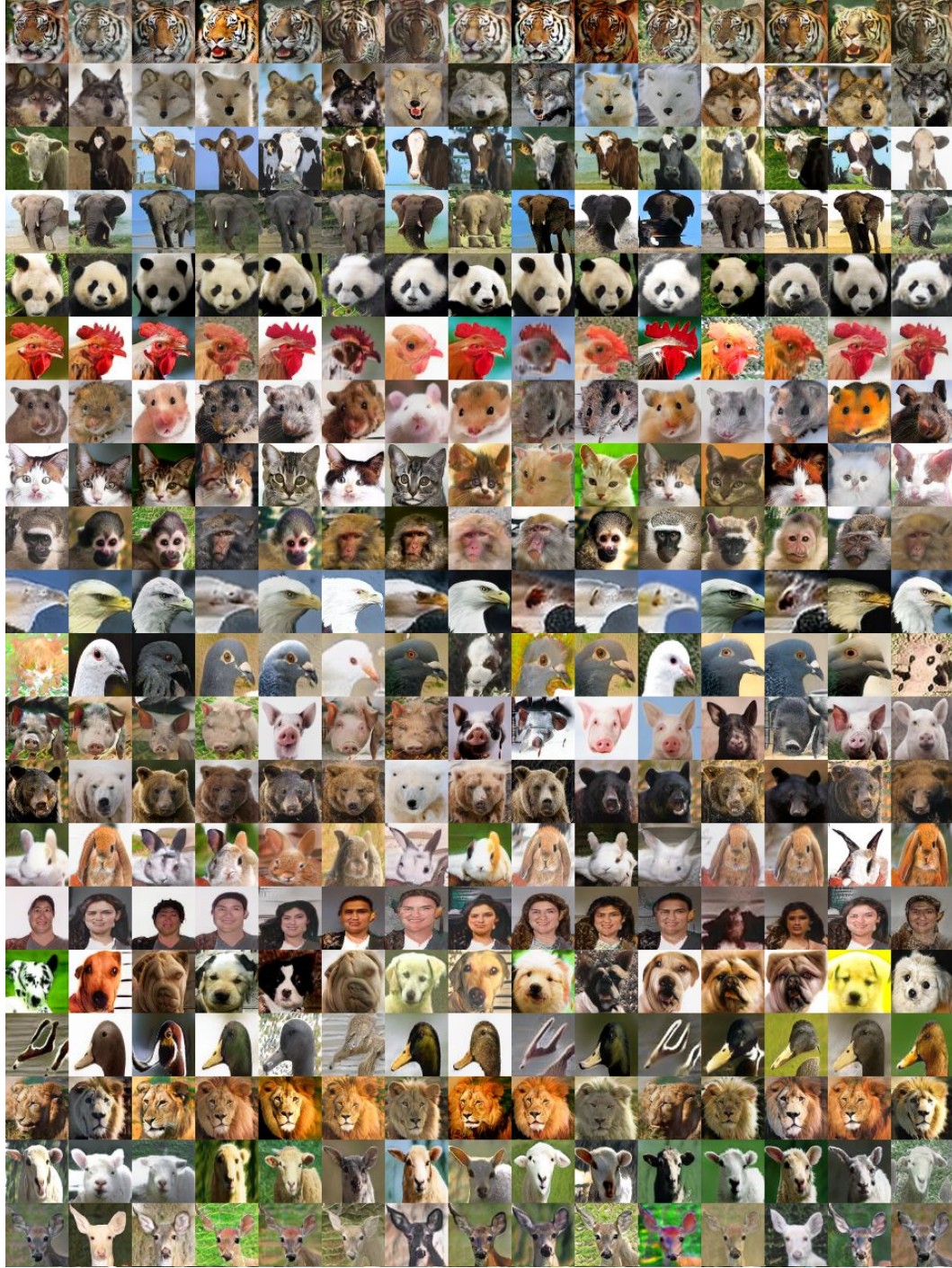

Figure 12: Randomly-generated images using the proposed method trained on the AnimalFace dataset. Each row represents a different class. The FID score is 16.

