# OpenReview forum: "Collapse by Conditioning: Training Class-conditional GANs with Limited Data"
_ICLR.cc/2022/Conference — ICLR 2022 Poster_

### Official Review · Reviewer_juAa · 2021-10-28

**Correctness:** 3
**Technical Novelty And Significance:** 2
**Empirical Novelty And Significance:** 3
**Recommendation:** 6
**Confidence:** 4

**Main Review:**

Strengths:
This paper is easy to follow and understand. In my opinion, the authors' observation and proposed training strategy are reasonable. Experiments on four datasets demonstrated the superiority of the proposed strategy.

Weaknesses:
1) In the paper, the authors claim that "the class-conditioning training is more robust to data volume reduction" is an intuitive belief. Based on such intuition, one contribution of this paper is pointing that cGANs are easier to suffer from mode collapse. However, I think this discovery may be widely known by the GAN community. Usually, a stronger condition would lead to worse diversity. For example, conditioning on semantic segmentations would largely decrease the generative diversity.

2) Although the solution/proposed strategy is simple and useful, some principles are not discussed or explored very well. According to experiments, unconditional training seems to correspond to better diversity yet worse quality, conditional training is the opposite. The proposed training seems to find a balance between them. However, is there always the best balance point and how can we arrive there? I believe that function $\lambda_t$ will largely influence training. For example, if we train cGANs with a large ratio and a relatively long time (e.g., increase $T_e$), will the trained generator collapse again?

**Summary Of The Paper:**

In this paper, the authors work towards training conditional GANs with limited data. Based on the observation that conditional GAN training suffers worse mode collapse than unconditional training, the authors proposed a training strategy that gradually injects conditional information into unconditional training. In other words, a learning curriculum that gradually replaces unconditional GANs training (losses and structure) with conditional ones is proposed.

Overall, the proposed learning curriculum is reasonable and achieved promising performance on data-efficient cGANs training.

**Summary Of The Review:**

Overall, the authors work on an interesting topic and present promising results.  In my opinion, the solution seems reasonable, yet there are still many problems that should be further explored. I highly suggest the authors further improve this work. However, I may not champion the current version.

---

> ### Author Response · Authors · 2021-11-20
> **Response to Reviewer juAa**
>
> We thank the reviewer for the constructive and valuable feedback.
>
> ### **Q1:** Strong conditioning is known to reduce diversity in GANs
>
> We agree with the reviewer that a decrease in diversity is observed in tasks with strong pixel-level conditioning, such as conditioning on semantic segmentation or on images (e.g. super-resolution).
> However, we do not believe that these observations directly translate to the class-conditional settings.
> On the contrary, the analysis performed in Fig. 2 shows that the class-conditional model achieves consistently better FIDs than the unconditional one if enough data is available (>5k images). This behavior could therefore be expected to continue when gradually reducing the available training data. In fact, as mentioned in section 2, some existing works investigate exploiting label supervision to improve unconditional GANs (referenced as "Salimans et al.", "Zhou et al.", and "Kavalerov et al." in the paper). We added this discussion to the revised Appendix (Section A.7), and we will include it in the final revision.
>
>
> ### **Q2:** Balancing between unconditional and conditional training
>
> We regret that the lack of some details of the transition process might have led to a misunderstanding. Therefore, we added a visualization of the transition function in Fig. 7 of Appendix (in the revised version). As shown, $T_s$ is the starting time and $T_e$ is the end time of the transition. Note that the end of the training is different from the end of the transition. We have added a new term $T_m$ to Fig. 7 to represent the maximum training iterations. After the end of the transition ($T_e$), the transition function $\lambda_t$ remains at its maximum value of 1. In other words, the weight of each loss (of Equation 3 in the paper) does not get adjusted anymore. The end of transition $T_e$ happens at about half the total training time $T_m$ in our experiments.
>
> Now, we would like to address the reviewer's concern in two parts:
>
> **Q2a) Balance between unconditional and conditional training.**
>
> Our method is not using $\lambda_t$ as a balancing weight between the unconditional and conditional objectives. $\lambda_t$ is used for a smooth transition to the conditional model. The value of $\lambda_t$ linearly increases from 0 at $T_s$ to 1 at $T_e$ and remains there for the rest of the training.
>
> **Q2b) Sensitivity to the settings of the transition function $\lambda_t$.**
>
> The proposed transition function has two hyper-parameters $T_s$ and $T_e$ (Equation 1 in the paper). In the ablation study of the paper (Table 3 and 4) we have shown that our approach is robust to even big variations in $T_s$ and $T_e$. Therefore, our method is not sensitive to the choice of the introduced hyper-parameters. More specifically, regarding longer transition time (bigger $T_e$), we provide two more results for $T_e=6k$ and $T_e=7k$ as asked by Reviewer iwi9 question Q6, further demonstrating the stability of our method with larger $T_e$. Regarding the end of training $T_m$, as mentioned in the response to Q5 of Reviewer iwi9, we train our experiments for maximum iterations of 9k, which is much longer than the end of the transition. The FID curves in Fig. 1 of the paper show that the method is able to remain stable, even though it is trained for a relatively large $T_m$.

---

> > ### Comment · Reviewer_juAa · 2021-11-25
> > **Additional Questions**
> >
> > I appreciate the authors' response, which helps me to better understand their training strategy. However, my concerns may not be fully solved.
> >
> > First, I am still confused about the principle of the proposed transition function, i.e. why it helps to solve mode collapse issue and improve the generation performance, and the most important: when will the transition training achieve the best performance. From the authors' answer to iwi9-Q5, the selected final model is the best model after the end of the transition, yet not the model at the end of training (9k training iterations). Meanwhile, as shown in Fig.1, the propose method may achieve the similar performance as unconditional training (Food 101).
> >
> > In addition, I wonder to know, how many images are used to test the FID score? Although the FID score is a popular metric for testing image generation quality, it may still be hard to reflect the diversity and quality that I mentioned in my comments.
> >
> > Overall, my main concern is still the analysis and discussion related to the proposed transition training strategy. From current experimental results, it indeed works. However, its working mechanism still unclear to me. I hope that the authors could further explain or explore it, and show us some detailed discussions.

---

> > > ### Author Response · Authors · 2021-11-27
> > > **Response to Reviewer juAa - More discussion on the proposed method**
> > >
> > > We thank the Reviewer for the additional feedback and provide answers below.
> > >
> > > ### **Q1:** Discussion on why our approach improves the generation performance.
> > >
> > > Our transitional training approach is motivated by the observed conditional collapse, discussed in Section 2. That is, our aim is to leverage the stability of unconditional learning, which does not undergo mode-collapse for the same amount of training data. In the initial training stages, we therefore solely utilize unconditional training. This allows the model to reach a more stable state, already capable of generating reasonably diverse and good-quality images. The gradual introduction of conditioning into both generator and discriminator then allows the model to gain the control necessary for the conditional generation. The main reason **why** our approach is successful is thus that it avoids the irreversible mode collapse that happens in early stages of conditional training. We will further clarify these aspects in the paper.
> > >
> > >
> > > ### **Q2:** When will our transition training achieve the best performance?
> > >
> > > Our goal is to achieve a conditional GAN with the best possible image generation ability. To achieve class consistency, the model is selected after the end of the transition $T_e$ (please see Q3 and Q4 of Reviewer iwi9). As observed in our experiments (See FID curves in Fig. 1), the FID continues to improve after the end of the transition, due to the benefits of conditional learning (see also the response to Reviewer juAa, Q1 in the initial comments). As in other GAN training strategies, overfitting may occur when the network is trained for too long, as observed in Fig. 1, Food101. However, this is not problematic for our approach, as the goal of our training strategy is not to achieve the best FID at the very end of training, but at any time after the completion of the transition. Therefore, following the standard procedure (as in e.g. StyleGAN and StyleGAN2-ADA), we select the model with the best FID achieved during training (best possible generation ability), but after the end of transition $T_e$ (ensuring class consistency).
> > >
> > > Furthermore, please note that our aim is to outperform the conditional baseline model. However, as discussed in the answer to Reviewer tZA6 (Minor Q3), our approach also achieve an FID which is better than the unconditional model in our experiments.
> > >
> > >
> > >
> > > ### **Q3:** How many images are used to test the FID score?
> > >
> > > Following the standard practice, we use 50k generated images to calculate FID. For real images, all of the training samples are used. Please note that we have also provided KID scores, which are known to be a suitable measure for small data [Karras et al. 2020, Binkowski et al. 2018]. Furthermore, we have also included class-wise FID, whenever possible, in our previous answers to Reviewer iwi9 (Q3).
> > >
> > >
> > > ### **Q4:** FID might not reflect the diversity and quality.
> > >
> > > As requested by Reviewer iwi9 (Q4), we have also included precision/recall as a measure of quality/diversity in the revised Appendix (Tables 8 and 9). Our approach significantly outperforms the conditional baselines in terms of recall (diversity), while achieving a better or similar precision. For more details on the results, please refer to Section A.4 of Appendix.

---

> > > > ### Comment · Reviewer_juAa · 2021-11-28
> > > > **Final suggestion**
> > > >
> > > > Thanks for the authors' response. The rebuttal partly solved my concerns. I would like to increase my score to "marginally above the acceptance threshold".
> > > >
> > > > The reason to accept this paper. To my best knowledge, it is the first work that focuses on class-conditional GANs training with limited data. The authors proposed a simple and efficient transition training strategy to improve generation performance. Comprehensive experimental results are provided and demonstrate the proposed strategy indeed works in most situations.
> > > >
> > > > The reason to reject this paper. Although good observation and intuition are presented in this paper, it lacks in-depth exploration. As an ICLR paper, it may bring limited insight to the community. For example, why the proposed transition training strategy (i.e. linear increasing) is the best choice? When we train conditional GAN models, to balance diversity and quality, how can we adjust the proposed transition strategy? I cannot find answers from the current version and the authors' feedback.

---

### Official Review · Reviewer_iwi9 · 2021-11-02

**Correctness:** 3
**Technical Novelty And Significance:** 4
**Empirical Novelty And Significance:** 3
**Recommendation:** 6
**Confidence:** 4

**Main Review:**

Strength:
1. The observation is interesting. Indeed, the mode collapse problem happens more frequently in conditional GANs (including other conditional GANs such as image-to-image, tex-to-image generation).
2. The paper is well-written and easy to follow. The high-level idea and the technical details behind are well-explained. The modification that changes the concatenation to addition operation in the generator makes sense and enables the transition idea.

Concerns:
1. The transition design for the loss is unclear. 1) The goal is to transit the training from unconditional to conditional settings, 2) the "no transition" results in Table 2 shows that the combined loss harms the performance. I am wondering if the authors have tried the transition that the loss function in Equation 3 is set to $L = (1 - \lambda)L_{uc} + \lambda L_{c}$. In this case, the weight for the unconditional loss is zero after the transition.
2. How do the authors determine the subset of classes and images to be used for the experiments?
3. How do the authors compute the FID and KID scores? Are these scores the average of the per-class ones? Per-class FID/KID scores can verify if the proposed method (unconditional training in the beginning) makes the generator to produce images that does not correspond to the input class label.
4. Since the authors claim to address the mode collapse issue under limited data setting for conditional GANs. The authors can consider to use the precision and recall scores to better measure the diversity of the generated images, especially the recall score.
Kynkäänniemi et al, "Improved Precision and Recall Metric for Assessing Generative Models," NeurIPS 2019.
5. What is the total training iteration used for all experiments? According to the paper, $T_s$ is set to 2K while $T_e$ is set to 4K. Is there any case that the best model used for computing the score is from the iteration before the transition ends?
6. Does the proposed harm the performance if $T_e$ is set to be close to the final training iteration? What are the results of setting $T_e$ to 6K or 7K in Table 4?

Suggestions (not related to rating):
1. The paper is easy to follow for people familiar with the conditional setting in StyleGAN2. However, the conditional setting is not well-introduced in the original paper which may confuses people when they read this paper. The authors can consider to briefly introduce how conditional generation is achieved in StyleGAN2 (both generator and loss sides) before describing the proposed method.
2. The authors can consider to use "clean FID" score due to the robustness.
Parmar et al., "On Buggy Resizing Libraries and Surprising Subtleties in FID Calculation."

**Summary Of The Paper:**

This work address the limited training data issue for class-conditional GANs. Inspired by the observation that unconditional GANs produce more diverse results under the same setting, the authors propose a learning algorithm that transits the training of unconditional toward class-conditional settings. The authors report FID and KID scores on four datasets to verify the proposed approach.

**Summary Of The Review:**

Overall, this is a good paper that addresses the limited data issue for conditional GANs. However, I have two major concerns: the transition design for the loss, and evaluation metrics. Please see Weakness section in the main review.

---

> ### Author Response · Authors · 2021-11-20
> **Response to Reviewer iwi9**
>
> We thank this reviewer for the thorough and constructive feedback.
>
> ### Q1: Transitioning of the form $L=(1-\lambda_t)L_{uc}+\lambda_tL_c$?
>
> Interestingly, we initially tried the loss function suggested by the reviewer. However, although competitive, our experiments yield better results with the loss formulation proposed in the paper. The results for the two formulations are as follows:
>
> Loss | ImageNet Carnivores || Food101 || CUB-200-2011 || AnimalFace ||
> --- | --- || --- || --- || --- ||
>  | FID | KID | FID | KID | FID | KID | FID | KID
> $L = (1-\lambda_t) L_{uc} + \lambda_t L_c$ | 18 | 0.0047 | 25 | 0.0088 | 25 | 0.0064 | 20 | 0.0035
> $L_{uc} + \lambda_t  L_c$ (Ours) | **14** |  **0.0021** |  **20** |  **0.0045**  | **22**  |  **0.0032**  |  **16** |  **0.0018**
>
> Based on the results, the unconditional and conditional training are not conflicting in the later stages of the training. Instead, having the unconditional loss actually helps the performance. However, as shown in Table 2 of the paper, and as pointed out by the reviewer, the two losses seem to be conflicting at the beginning of the training, resulting in a bad performance in the absence of the transition. We have added this ablation to Appendix in the revised version (Section A.6, Table 10).
>
> ### Q2: How the subset of classes and images were determined.
>
> We use a random selection of the classes and images for all datasets (see section 4.1).
>
> ### Q3: FID and KID calculation.
>
> Following the common practice (StyleGAN, BigGAN, etc.), we calculate the FID and KID metrics unconditionally, with a class sampling distribution that matches the class distribution of the training dataset. We could not provide the class-wise FID and KID due to the insufficient class-wise sample size. Here, we report the class-wise metrics for ImageNet Carnivores and Food101, by using the full number of images for corresponding classes, by including the additional images not used for training. Results are also added to the revised appendix (Table 7).
>
>
> Method | ImageNet Carnivores || Food101 ||
> --- | --- || --- ||
>  | FID | KID | FID | KID
> C-StyleGAN+ADA | 139 | 0.179 | 100 | 0.079
> C-StyleGAN2+ADA+ProjD | 151 | 0.199 | 97 | 0.081
> C-StyleGAN2+ADA+Lecam | 90 | 0.096 | 56 | 0.027
> Ours | **30** |  **0.011** |  **44**  |  **0.019**
>
> Note that the values are generally larger for all methods since a large fraction of FID/KID reference set was not used for training. However, the relative values are still consistent with the metrics reported in the paper. These measures, along with the generated images in the appendix (Figs. 9-12), show the class consistency of the proposed method.
>
> ### Q4: Precision and recall
>
> Here we provide the precision and recall proposed by Kynkäänniemi et al., with the implementation provided by StyleGAN2+ADA (also added as Tables 8 and 9 in the revised Appendix). As shown, unconditional training always yields a higher recall, as it can generate between-mode images (bigger diversity). Among the conditional methods, ours yields significantly better recall, as well as comparable precision. Low recall values for the conditional baselines confirm the observed mode collapse.
>
> Method | ImageNet Carn. || Food101 || CUB-200-2011 || AnimalFace ||
> --- | --- || --- || --- || --- ||
> | P | R | P | R | P | R | P | R
> UC-StyleGAN2+ADA | 0.77 | **0.300** | 0.710 |**0.187** | 0.70 | **0.232** | 0.76 | **0.430**
> C-StyleGAN2+ADA+ProjD | 0.81 | 0.0 | 0.74 | 0.002 | 0.76 | 0.067 | **0.83** | 0.0
> C-StyleGAN2+ADA+Lecam | 0.81 | 0.046 | 0.73 | 0.004 | 0.76 | 0.097 | **0.83** | 0.0005
> Ours | **0.82** |  **0.263** |  **0.82** | **0.068** | **0.77** | **0.229**  | **0.83** |  **0.314**
>
> **Class-wise Metrics:**
>
> Method | ImageNet Carn.|| Food101 ||
> --- | --- || --- ||
> | P | R | P | R
> C-StyleGAN2+ADA | **0.75** | 0 | 0.75 | 0
> C-StyleGAN2+ADA+ProjD | 0.68 | 0.0 | **0.78** | 0.00005
> C-StyleGAN2+ADA+Lecam | 0.67 | 0.0001 | 0.70 | 0.0148
> Ours | 0.73 | **0.1205** | 0.64 | **0.1049**
>
> ### Q5: Model selection and the total iterations?
>
> We use a maximum of 9k training iterations. The best models are selected only after the end of the transition ($T_e=4$k): ImageNet Carnivores at 6451, Food101 at 6652, CUB-200-2011 at 4233, AnimalFace at 8265. We will clarify this in the revised version.
>
> ### Q6: Results for larger $T_e$ (6K & 7K) in Table 4?
>
> The completed analysis of the importance of the transition ending time $T_e$, including 6k and 7k, is shown below:
>
> Experiment | ImageNet Carnivores || Food101 ||
> --- | --- || --- ||
> $T_e$| FID | KID | FID | KID
> 3k | 21 | 0.0053 | 16 | 0.0032
> 4k | 20 | 0.0045 | 14 | 0.0021
> 5k | 23 | 0.0062 | 15 | 0.0029
> 6k | 24 | 0.0075 | 15 | 0.0028
> 7k | 22 | 0.0065 | 14 | 0.0021
>
> Results show that changing the hyperparameters only creates slight fluctuations in the best FID. The method is able to achieve a very good FID in all the setups with different hyperparameters.
>
> **Suggestions 1 and 2:** We will carefully consider the reviewer's suggestions for the final version

---

> > ### Comment · Reviewer_iwi9 · 2021-11-23
> > **Response to authors**
> >
> > Thank the authors for the thorough responses. Although it would be great if the authors can provide more intuition/analysis to the observation described in Q1, all my concerns are addressed. Overall it is a good paper, and I hope the authors can release the code to stimulate future research.

---

> > > ### Author Response · Authors · 2021-11-27
> > > **Reviewer iwi9 - The release of the code**
> > >
> > > We are glad that the reviewer's concerns are addressed. We provided the code in the general comment above.

---

### Official Review · Reviewer_F98y · 2021-11-02

**Correctness:** 4
**Technical Novelty And Significance:** 2
**Empirical Novelty And Significance:** 3
**Recommendation:** 8
**Confidence:** 4

**Main Review:**

Strengths:

[S1] The authors provide interesting observations of cGANs vs unconditional GANs in limited data settings which have not been observed or reported in previous works.

[S2] Comprehensive experiments are conducted to demonstrate the claim that class-conditioning causes mode collapse in limited data settings, whereas unconditional learning leads to satisfactory generative ability.

[S3] The authors propose a simple yet effective method for training cGANs that effectively prevents the observed mode-collapse and results in significant empirical improvements compared to state-of-the-art methods and established baselines.

[S4] The authors also perform a comprehensive ablation analysis over different components of their method.


Weaknesses:

[W1] The authors incorporate their method only in StyleGAN2 with ADA. While I understand that it is a recent state-of-the-art method for unconditional and class conditional image generation under a limited-data setup, it would be great if authors can verify the efficacy of their approach when applied to different models like BigGAN, etc.

[W2] The authors do not compare with some of the datasets that a recent method uses - Regularizing Generative Adversarial Networks under Limited Data (Tseng et al. 2021). Tseng et al. also perform experiments with GANs where they whittle down training data and they primarily conduct experiments with CIFAR10 and CIFAR100. The authors do not show their observations and experiments on C10 and C100. It would be nice to have results on these two datasets.

[W3 minor] The technical novelty of the paper is not much. Similar strategies of transitioning from one objective to include another objective can be seen often. For example, training a model using a pre-text task (contrastive learning, solving jigsaw, etc.) and then finetuning for the target task. Authors follow a similar approach where they kind of pretrain using unconditional learning and later include conditional learning. Having said that, the observations made as well as the application of this simple trick to mitigate mode collapse in cGANs are quite novel (empirically).


**Summary Of The Paper:**

This work studies the problem of training class-conditional GANs in limited data settings. The authors first empirically demonstrate that class-conditioning results in mode collapse in a limited data regime whereas unconditional learning provided satisfactory generative ability. They perform a comprehensive analysis by gradually reducing the size of the training set in two ways, a) reducing the number of classes while keeping a number of 100 training images per class and b) reducing the number of images per class while using 50 classes in all cases. In both cases, the conditional GAN achieves a better FID for larger datasets while performance deteriorates significantly (experiencing mode collapse) when the dataset size is reduced. On the other hand, the unconditional model achieves consistently better FID in limited data settings. Based on this observation, the authors propose a method of injecting the class conditioning by transitioning from unconditional to the conditional case, in an incremental manner. They delineate the proposed architectural changes and training objectives. Authors base their experiments on StyleGAN2 with adaptive data augmentation (ADA) and four datasets. The empirical results suggest significant improvements compared to state-of-the-art methods and established baselines. The authors also perform a comprehensive ablation analysis to understand the contribution of different components.

**Summary Of The Review:**

The authors discover an interesting behavior of cGANs in limited data settings and propose a simple yet very effective method to solve the problem. The authors perform a comprehensive analysis of the problem followed by a comprehensive set of experiments and ablation analysis showing the efficacy of their method. Overall, the paper seems to make important observations and empirical contributions.

---

> ### Author Response · Authors · 2021-11-20
> **Response to Reviewer F98y**
>
> We thank the reviewer for the very positive and valuable feedback.
>
> ### **Q1**: Other architectures, such as BigGAN and ContraGAN
>
> We here restate the answer to reviewer tZA6, who asked the same question in Q2. The observation and the proposed method are based on setups where unconditional training is stable. BigGAN and BigGAN-based models like ContraGAN are designed and optimized (architecture, hyper-parameters, etc.) only for conditional training. How to transform BigGAN into a capable unconditional architecture has not been explored. We tried to follow the strategy described in the ADA paper [Karras et al. 2020] to design an unconditional BigGAN baseline. However, similar to what [Karras et al. 2020] have reported (Fig. 19 of Appendix in their paper), we were not able to find a stable configuration of unconditional BigGAN in the low-data regime. We therefore also conduct our analysis on the StyleGAN2 architecture. Note that StyleGAN2+ADA represents the state-of-the-art for both conditional and unconditional generation in the low-data regime. It is therefore the strongest and most relevant baseline for our study. We have added this discussion to the revised Appendix (Section A.8), and we will include it in the main paper in the final revision.
>
> References:
> Karras et al., Training Generative Adversarial Networks with Limited Data, NeurIPS, 2020.
>
>
> ###  **Q2**: Experiments on CIFAR.
>
> Indeed providing similar investigations on more datasets can further benefit our paper. However, we hope the extensive experiments and ablation on the four datasets provided in the paper are already sufficient to show the validity of our observation, as well as the efficacy of the proposed solution. Nevertheless, as suggested by the reviewer, we conducted extensive experiments on CIFAR100. Results for 4 different settings (number of classes and images) are provided in the appendix of the revised version (Section A.2, Table 6), as well as in the table below. We observe a similar behavior on CIFAR100, where our approach outperforms the conditional baseline (C-StyleGAN+ADA), achieving FID and KID better or on-par with the unconditional baseline (UC-StyleGAN+ADA).
>
> Num. (Classes, Images/class) | (20, 500) || (50, 50) || (50, 300) || (100, 300) ||
> ---|---||---||---||---||
> Method | FID | KID | FID | KID | FID| KID | FID | KID
> UC-StyleGAN+ADA  | **7** |  **0.0006** | **20** | 0.0022 | **6** | **0.0008**  | **6** | 0.0021
> C-StyleGAN+ADA | 12 | 0.0033 | 23 | 0.0036 | 9 | 0.0025 | 13 | 0.0053
> Ours | **7** | **0.0006** | **20** | **0.0014** | **6**  |  **0.0008**  |  **6**  |  **0.0012**
>
> ### **Q3**: Technical novelty
>
> We would like to stress that our first main contribution is the extensive analysis of the behavior of conditional and unconditional models in the low-data regime. Our second main contribution is transitional training, which reduces the data required for learning a conditional model. Our aim is to propose a method that is as simple, yet effective, as possible in order to ease the integration and adaption by researchers and practitioners in the field.

---

> > ### Comment · Reviewer_F98y · 2021-11-27
> > **Response to Authors**
> >
> > I would like to thank the authors for their responses and for addressing my concerns. Overall it is a very good pape.

---

### Official Review · Reviewer_tZA6 · 2021-11-02

**Correctness:** 3
**Technical Novelty And Significance:** 3
**Empirical Novelty And Significance:** 4
**Recommendation:** 5
**Confidence:** 4

**Main Review:**

Strengths:
- The writing of this paper is easy to follow.
- The idea is novel and intuitive. The authors propose a new training strategy to transfer stylegan2-ada from an unconditional to a conditional approach. And in the results, the best FID was obtained.

Weaknesses & Questions:
- The reason why this strategy only requires limited data has not been analyzed. Is it the stylegan-ada that reduces the need for data? If so, the "ada" strategy is not a contribution of this paper. As the authors claim: " The proposed method for training cGANs with limited data results not only in stable training…" . More experiments should be conducted to verify that the proposed method can reduce the need for data. For example, applying the proposed method to stylegan or stylegan2 and training on the same data.
- Are other architectures supported? The results provided are only implemented on stylegan-ada. However, these statements in Sections 1 and 2 may lead the reader to believe that the proposed approach is a general approach that can be applied to different cGANs architectures. Therefore, in addition to the results of stylegan2-ada, implementations on different architectures should be provided. For example, BigGAN, ContraGAN, etc.
- More qualitative results are compared. Tables 1 to 4 and Figure 2 provide quantitative results at different settings, however, providing more qualitative results at these settings can help to more visually assess the enhancements from the method.

Minors:
- Details of L^D_uc, etc. should be specified, and are there additional hyper-parameters for each term?
- For better illustration, Ts and Te can be added to Figure 1 (first row), although the ranges of Ts and Te can be easily guessed.
- In Figures 1 and 2, "our" outperforms the unconditional version in the FID. Why is the Fid further reduced compared to the unconditional version?



**Summary Of The Paper:**

In this paper, authors proposed a new training strategy which transfer the StyleGAN2-ada to the conditional version gradually by injecting conditional information into the generator and the objective function during the training phase. The proposed method is capable of training on limited data and generating high-quality images. However, the strategy of this method is only applicable to StyleGAN-like architecture and experiments can be improved.

**Summary Of The Review:**

This paper provides an intuitive solution to move stylegan2-ada from an unconditional to a conditional approach and achieve the best performance on the FID metric, however more experiments, as listed above, should be given to help a better evaluation of all aspects of this work.

---

> ### Author Response · Authors · 2021-11-20
> **Response to Reviewer tZA6**
>
> We thank the reviewer for their valuable feedback. We are glad that the reviewer finds the proposed method novel and intuitive, and the structure of the paper easy to follow.  In the following, we will address the question about ADA by providing more results. Then we discuss our choice of architecture, and finally, provide more visual results as requested.
>
>
> ### **Q1.a:** The reason why our approach is able to cope with limited data:
>
> As discussed in Sections 2 and 3.1 of the paper, the proposed method is based on the observation that unconditional training of GANs can be stable in low-data regimes where conditional training fails. This allows us to exploit unconditional training to improve the conditional model in the limited data setup.
>
>
> ### **Q1.b:** Is it ADA that reduces the need for data in our approach?
>
> Please note that we employ ADA for all models and baselines in the paper, including both the conditional baseline "C-StyleGAN2" and our approach "Ours" in Table 1 of the paper. Therefore, the data reduction achieved by our approach is complementary to ADA.
>
> ### **Q1.c:** Experiments without ADA.
>
> As requested by the reviewer, we performed experiments to verify our observation and the effectiveness of our approach, when not employing ADA. We use the ImageNet carnivores dataset. Due to the absence of ADA, the unconditional model needs a somewhat larger dataset to converge. We use 50 classes and 500 images/class. The unconditional "UC-StyleGAN2", baseline conditional "C-StyleGAN2", and "Ours" are trained on this same data. The results are given in the table below.
>
> Method | FID | KID
> --- | --- | ---
> UC-StyleGAN2 | 13 | 0.0067
> C-StyleGAN2 |  99 | 0.0795
> Ours |  **8** | **0.0020**
>
> As the FID and KID scores show, the observed conditioning collapse happens even in the absence of ADA. Our method is still able to solve the problem by leveraging a stable behavior of unconditional training. We have also updated Appendix with these results along with FID curves (Section A1: Fig. 5, and Table 5).
>
> ### **Q2:** Other architectures, such as BigGAN and ContraGAN
>
> The observation and the proposed method are based on setups where unconditional training is stable. BigGAN and BigGAN-based models like ContraGAN are designed and optimized (architecture, hyper-parameters, etc.) only for conditional training. How to transform BigGAN into a capable unconditional architecture has not been explored. We tried to follow the strategy described in the ADA paper [Karras et al. 2020] to design an unconditional BigGAN baseline. However, similar to what [Karras et al. 2020] have reported (Fig. 19 of Appendix in their paper), we were not able to find a stable configuration of unconditional BigGAN in the low-data regime. We therefore also conduct our analysis on the StyleGAN2 architecture. Note that StyleGAN2+ADA represents the state-of-the-art for both conditional and unconditional generation in the low-data regime. It is therefore the strongest and most relevant baseline for our study. We have added this discussion to the revised Appendix (Section A.8), and we will include it in the main paper in the final revision.
>
> References:
> Karras et al., Training Generative Adversarial Networks with Limited Data, NeurIPS, 2020.
>
> ### **Q3:** More qualitative results.
>
> In addition to the qualitative results in Fig. 1, generated samples of our method on all 4 datasets are provided in the appendix (Figs. 9-12). Following the reviewer's suggestion, we have added more visual results for comparison with other baselines in Fig. 8 in Appendix.
>
> ### **Minor Q1:** Details of the loss functions.
>
> The unconditional and conditional loss functions are exactly as those of StyleGAN2, which consist of the adversarial loss (i.e. the non-saturating logistic loss), as well as the R1 loss and path length regularization. We do not alter the hyper-parameters of the baselines and use the automatic hyper-parameter selection provided in the StyleGAN2+ADA implementation. We will add the details of the baseline loss functions to the implementation details of the paper.
>
> ### **Minor Q2:** Adding $T_s$ and $T_e$ to the plots in Fig. 1.
>
> We thank the reviewer for this idea and will incorporate it into the final version. Note that these values are provided in the implementation details in Section 4.1.
>
> ### **Minor Q3:** Why does our method outperform unconditional training in terms of FID?
>
> We believe that this is another interesting discovery of our study. That is, our transitional learning strategy achieves results better than both the standard unconditional and conditional training. We believe this is due to the potential benefits of conditional training. Note that our analysis in Fig. 2 shows that conditional training achieves better FID than unconditional if enough data is available. Our approach is thereby able to bring this benefit to the low-data regime, allowing the model to surpass even the unconditional baseline.

---

> > ### Comment · Reviewer_tZA6 · 2021-11-30
> > **Response to Authors**
> >
> > Thanks for the authors' response and most of my concerns have been addressed.

---

### Author Response · Authors · 2021-11-20
**We thank all reviewers for their positive and valuable feedback**

We are glad that the reviewers find the observation made in the paper novel and important (Reviewer F98y), as well as interesting (Reviewer iwi9). The reviewers also find the idea in the paper novel and intuitive (Reviewer tZA6), and the method simple and effective (Reviewer F98y). We are also delighted that the reviewers found the experiments and analysis to be comprehensive (Reviewer F98y), and that they demonstrate the efficacy (Reviewer F98y) and superiority (Reviewer juAa) of the proposed method, as well as significant improvements compared to state-of-the-art (Reviewer F98y). The reviewers also point out that the paper is well-written and easy to follow (Reviewers tZA6, iwi9, and juAa). Below, we address all reviewers' questions and concerns.

---

### Decision · Program_Chairs · 2022-01-20

**Decision:**

Accept (Poster)

**Comment:**

This paper examines conditional GANs, which are found to lead to model collapse in low data settings. The paper proposes what appears to be a simple but effective method that addresses the issue. Reviewers were generally happy with the experiments and the utility of the observations and analysis. Code for the method was provided during the author response period. Only one reviewer did not vote to accept this paper, but they did acknowledge that the authors had addressed their concerns during the discussions with the authors. All others rated the paper as an accept.
The AC recommends accepting this paper.